# The Long Non-Coding RNA *HOXA-AS2* Promotes Proliferation of Glioma Stem Cells and Modulates Their Inflammation Pathway Mainly through Post-Transcriptional Regulation

**DOI:** 10.3390/ijms23094743

**Published:** 2022-04-25

**Authors:** Elisa Le Boiteux, Pierre-Olivier Guichet, Konstantin Masliantsev, Bertille Montibus, Catherine Vaurs-Barriere, Céline Gonthier-Gueret, Emmanuel Chautard, Pierre Verrelle, Lucie Karayan-Tapon, Anne Fogli, Franck Court, Philippe Arnaud

**Affiliations:** 1Université Clermont Auvergne, CNRS, Inserm, GReD, F-63000 Clermont-Ferrand, France; elisalb@bmb.sdu.dk (E.L.B.); bertille.montibus@kcl.ac.uk (B.M.); catherine.barriere@uca.fr (C.V.-B.); celine.gonthier_gueret@uca.fr (C.G.-G.); anne.fogli@uca.fr (A.F.); 2ProDiCeT UR 24144, Université de Poitiers, F-86000 Poitiers, France; pierre-olivier.guichet@inserm.fr (P.-O.G.); konstantin.masliantsev@univ-poitiers.fr (K.M.); lucie.karayan-tapon@chu-poitiers.fr (L.K.-T.); 3Laboratoire de Cancérologie Biologique, CHU de Poitiers, F-86000 Poitiers, France; 4Pathology Department, Jean Perrin Center, F-63000 Clermont-Ferrand, France; emmanuel.chautard@clermont.unicancer.fr; 5INSERM, U1240 IMoST, Université Clermont Auvergne, F-63000 Clermont-Ferrand, France; 6CIMB, INSERM U1196 CNRS UMR9187, Curie Institute, F-91400 Orsay, France; pierre.verrelle@curie.fr; 7Radiotherapy Department, Curie Institute, F-75248 Paris, France; 8CNRS UMR 9187, INSERM U1196, Institut Curie, PSL Research University and Paris-Saclay University, F-91405 Orsay, France; 9Radiation Oncology Department, Institut Curie, F-75005 Paris, France

**Keywords:** cancer, glioma stem cells, long non-coding RNA, HOX genes, post-transcriptional regulation, histone marks

## Abstract

Glioblastomas represent approximatively half of all gliomas and are the most deadly and aggressive form. Their therapeutic resistance and tumor relapse rely on a subpopulation of cells that are called Glioma Stem Cells (GSCs). Here, we investigated the role of the long non-coding RNA *HOXA-AS2* in GSC biology using descriptive and functional analyses of glioma samples classified according to their isocitrate dehydrogenase (*IDH*) gene mutation status, and of GSC lines. We found that *HOXA-AS2* is overexpressed only in aggressive (*IDHwt*) glioma and GSC lines. ShRNA-based depletion of *HOXA-AS2* in GSCs decreased cell proliferation and altered the expression of several hundreds of genes. Integrative analysis revealed that these expression changes were not associated with changes in DNA methylation or chromatin signatures at the promoter of the majority of genes deregulated following *HOXA-AS2* silencing in GSCs, suggesting a post-transcriptional regulation. In addition, transcription factor binding motif enrichment and correlation analyses indicated that *HOXA-AS2* affects, directly or indirectly, the expression of key transcription factors implicated in GCS biology, including E2F8, E2F1, STAT1, and ATF3, thus contributing to GCS aggressiveness by promoting their proliferation and modulating the inflammation pathway.

## 1. Introduction

Glioma is the most common primary malignant brain tumor, affecting more than 200,000 individuals worldwide each year [1]. The 2007 World Health Organization (WHO) classification, based on histopathological features, distinguished four glioma grades (I–IV). Grade IV or glioblastoma (GBM) is the most aggressive form [2]. The last WHO classification, released in 2021, also takes into account molecular features, specifically the isocitrate dehydrogenase (IDH) gene mutation status [3]. Aggressive gliomas, which include GBM, harbor wild-type *IDH1* and *IDH2* genes (*IDHwt*), whereas gliomas with better prognosis carry mutated *IDH1* and *IDH2* genes (*IDHmut*) [4]. Although their prevalence is relatively low, GBM is a major cause of morbidity and mortality, and display very low two- and five-year survival rates (1). Despite the aggressive treatment (surgical resection followed by chemotherapy and/or radiotherapy), GBMs present a very high recurrence rate, and the median survival time after diagnosis does not exceed 18 months. It has been proposed that therapeutic resistance and tumor relapse rely on a tumor cell subpopulation with stem cell characteristics, called glioma stem cells (GSCs) [5,6]. Therefore, determining the molecular bases of the GSC pathological potential is a prerequisite to improve *IDHwt* glioma/GBM management.

In a recent study, we showed that widespread HOX gene overexpression is a molecular signature of *IDHwt* glioma samples and GSCs [7]. In humans, the 39 HOX genes are grouped in four genomic clusters (HOXA, B, C, and D) in chromosomes 7, 17, 12, and 2, respectively. The members of this evolutionarily conserved family encode homeodomain transcription factors that are critical for normal development [8]. When deregulated, they may be implicated in the GSC tumorigenic potential [9]. Indeed, genetic manipulation of individual HOX genes in GBM cell lines showed that several of them, such as *HOXA5*, *A7*, *A10*, *D9*, and *D10*, can act as oncogenes by promoting cancer cell viability, invasion, and migration and/or by reducing cell death [8,10,11].

In addition to HOX genes, some of the 18 referenced non-coding antisense transcripts found at the four HOX clusters may also contribute to the GSC tumorigenic potential. For instance, HOX transcript antisense intergenic RNA (*HOTAIR*) and HOXA Transcript Antisense RNA Myeloid-Specific 1 (*HOTAIRM1*), which are located in the HOXC and HOXA loci, respectively, are strongly expressed in GBM where they have pro-tumor functions [12,13,14,15,16,17]. HOXA cluster antisense RNA 2 (*HOXA-AS2*) also emerge as an actor of glioma biology. This long non-coding RNA (lncRNA) is located in the HOXA cluster and is overexpressed in various cancer types, such as leukemia, gastric, hepatocellular, colorectal, breast, and gallbladder cancer [18,19,20,21,22,23,24], where it functions as an oncogene. Its overexpression has also been recently observed in malignant glioma [25]. A study performed in nonprimary glioblastoma cell lines showed that *HOXA-AS2* promotes their proliferation and invasion [25,26]. However, its precise function has not been investigated yet in primary GBM and in GSCs. Particularly, it is not known whether *HOXA-AS2* contributes to the oncogenic potential associated with aberrant HOX gene activation in GSCs. To address this important question, we characterized *HOXA-AS2* expression in *IDHwt* glioma samples and performed complementary descriptive and functional analyses in *IDHwt* GSC lines.

## 2. Results

### 2.1. HOXA-AS2 Is Specifically Expressed in IDHwt Glioma Samples and GSC Lines

As dozens of *HOXA-AS2* isoforms are predicted by the Gencode gene project, we first assessed the expression of all *HOXA-AS2* isoforms in three normal brain tissue samples (control), eight *IDHwt* and five *IDHmut* glioma samples, and two GSC lines using a strand-oriented RNA-seq approach. *HOXA-AS2* was not expressed in control and *IDHmut* samples. Conversely, we observed a robust signal for some *HOXA-AS2* isoforms in most, but not all, *IDHwt* samples and in the two GSC lines (Figure 1a). Refined analyses of one *IDHwt* glioma sample by 5′ and 3′ RACE-PCR (Appendix A), combined with analyses of the RNA-seq pattern, showed that the major *HOXA-AS2* transcript in glioma corresponded to a longer form of the ENST0000522193.1 isoform that initiates from a CpG island promoter (CGI 46) and contains at least two exons (Figure 1a and Appendix A). The rest of the study focused on this major isoform.

RT-qPCR analysis (*n* = 10 control samples, *n* = 8 *IDHmut* and *n* = 43 *IDHwt* glioma samples, and *n* = 6 GSC lines) confirmed that *HOXA-AS2* was expressed only in *IDHwt* glioma samples and GSC lines (Figure 1b), while it was virtually undetectable in control and *IDH**mut* glioma samples. To assess the reproducibility of these observations, we performed the same analyses in an independent cohort (“TCGA cohort”) that included 5 control, 415 *IDHm**ut*, and 134 *IDH**wt* glioma samples [27]. We could confirm that *HOXA-AS2* was expressed in *IDH**wt* samples, but not in control and *IDH**mut* glioma samples (Figure 1c). We observed a similar expression pattern for *HOTAIRM1* but not for *HOTAIR*, two HOX lncRNAs involved in glioma biology. Specifically, *HOTMAIR1* was expressed in both *IDHwt* and GSCs, and *HOTAIR* only in *IDHwt* samples. *HOTTIP*, another lncRNA, was almost undetectable or extremely weakly expressed in all samples (Appendix A). Altogether, these analyses showed that *HOXA-AS2* is overexpressed in aggressive *IDHwt* glioma samples and GSC lines.

Furthermore, Kaplan–Meier analyses of data from the R2: Genomics analysis and visualization platform (https://r2.amc.nl; accessed on 13 April 2022) and GDC data portal (https://portal.gdc.cancer.gov/ accessed on 13 April 2022) showed that *IDHwt* samples with a higher *HOXA-AS2* expression level tend to have a poorer survival outcome (Appendix A).

### 2.2. HOXA-AS2 Is a Nuclear RNA and Its Overexpression in IDHwt Glioma Is Associated with H3K27me3 Loss at Its Promoter

To characterize *HOXA-AS2* expression, we first determined the relative distribution of its mature spliced transcript in cell compartments. Expression analysis in three GSC lines indicated that it was localized in the cytoplasmic and nuclear compartments (Figure 2a).

Next, we investigated the molecular bases of *HOXA-AS2* overexpression in *IDHwt* glioma. We assessed the DNA methylation status of the *HOXA-AS2* transcript in control (*n* = 8) and *IDHwt* (*n* = 55) samples by Infinium HumanMethylation450 (HM450K) BeadChip Arrays. In agreement with our previous observation made at the four HOX clusters [7], DNA methylation was increased at the *HOXA-AS2* locus in *IDHwt* samples compared with the healthy control. However, this gain was localized mainly in the transcribed region, while methylation remained low at the promoter region. This indicated that *HOXA-AS2* overexpression was not associated with changes in its promoter methylation status (Figure 2b). Conversely, ChIP analysis of the *HOXA-AS2* promoter showed a marked decrease in the repressive H3K27me3 mark associated with a gain in the permissive H3K4me3 and H3K9ac marks in *IDHwt* glioma (*n* = 7) compared with control samples (*n* = 5) (Figure 2c).

To evaluate the relative contribution of histone modifications and DNA methylation changes to *HOXA-AS2* overexpression also in GSCs, we analyzed two *IDHwt* GSC lines (GSC-6 and GSC-11) and a neural stem cell line (H9-NSC; control) using ChIP-seq, RNA-seq, and Infinium Methylation EPIC BeadChips. *HOXA-AS2* expression in GSC-11, and, to a lesser extent, in GSC-6 cells, was associated with a gain in H3K4me3 in the promoter region and in DNA methylation (analyzed only in the GSC-11 line) in the rest of the locus, and with a marked decrease in H3K27me3 throughout the locus (compared with control H9-NSCs). This decrease was less important in GSG-6 than in GSC-11 cells, and mirrored the different *HOXA-AS2* expression levels in these two cell lines (Figure 2d). This pattern is in agreement with our previous observation that the H3K27me3 status recapitulates the transcriptional activity at HOX clusters [7]. Altogether, these observations suggest that H3K27me3 loss at the *HOXA-AS2* CGI/promoter is one the main mechanisms to explain its expression in GSC cells.

### 2.3. HOXA-AS2 Ectopic Overexpression Poorly Affects Human Neural Stem Cell Biology

To assess *HOXA-AS2* function, we first investigated the consequence of overexpressing spliced *HOXA-AS2* (longer form of the ENST0000522193.1 isoform) in human neural stem cells (H9-NSCs; two stable lines overexpressing *HOXA-AS2* and two stable lines containing the empty pcDNA 3.1 vector; each experimental-control couple was generated in an independent transfection experiment). We confirmed *HOXA-AS2* expression in the two *HOXA-AS2*-expressing H9-NSC lines and its absence in the two control lines (Appendix A).

We did not observe any difference in cell morphology, proliferation, and apoptosis in *HOX-AS2*-expressing and control H9-NSC lines (Figure 3a,b, Appendix A). Molecular characterization of the four cell lines did not highlight any significant DNA methylation difference (Infinium Methylation EPIC BeadChips) between *HOX-AS2*-expressing and control cell lines (Figure 3c). RNA-seq analysis revealed that the expression of only four genes was significantly altered in *HOX-AS2*-expressing cells, and *HOXA-AS2:* Fatty Acid Binding Protein 3 (*FABP3*) and Regulator of G Protein Signaling 16 (*RGS16*) were upregulated, while the endonuclease Schlafen Family Member 13 (*SLFN13*) and EBF transcription factor 2 (*EBF2*) were downregulated (Figure 3d). Interestingly, *FABP3* and *RGS16* have been proposed as markers of invasive glioma [28,29], and *EBF2* positively regulates neuronal migration [30].

These observations indicated that *HOXA-AS2* overexpression in healthy human neural stem cells does not lead to detectable phenotypic alterations, but alters the expression of a few genes that may be relevant for glioma biology.

### 2.4. HOXA-AS2 Knockdown Affects GSC Morphology

To assess its function in GSCs, we silenced *HOXA-AS2*, using an shRNA approach, in the GSC-6 and -11 lines that overexpress *HOXA-AS2,* although at different levels (Figure 2d). The transfection of two independent shRNAs that target exon 2 led to a decrease in *HOXA-AS2* expression by 50 to 70%, compared with control cells transfected with a nonsilencing shRNA (shNS) (Figure 4a,b). *HOXA-AS2* silencing altered neurosphere formation (a marker of aggressiveness) in both cell lines (Figure 4c). The quantification of cell viability at 2, 4, 7, and 9 days after *HOXA-AS2* silencing showed an important decrease in GSC proliferation compared with control cells (Figure 4d).

### 2.5. E2F-8 and -1 Are Candidate Factors to Mediate HOXA-AS2 Function in GSCs

Detailed analysis of the transcriptional landscape showed widespread transcriptional alterations following *HOXA-AS2* silencing. *HOXA-AS2* silencing led to the deregulation of 1725 genes (975 up- and 750 downregulated, respectively) in GSC-6 cells and of 784 genes (457 up- and 327 downregulated, respectively) in GSC-11 cells (Appendix A; Figure 5a). This difference between cell lines is consistent with the higher residual *HOXA-AS2* expression level in silenced GSG-6 than in GSC-11 cells (Figure 4b). Despite this difference in the number of affected genes, gene ontology analyses showed that the same pathways were affected in both GSC lines. The group of downregulated genes following *HOXA-AS2* silencing was enriched in genes involved in the cell cycle, specifically in cell and nuclear division. The upregulated group was enriched in genes involved in the inflammatory and immune response pathways (Figure 5b).

To determine whether this transcriptional alteration could be due to the initial alteration of a few “master” transcription factors, we analyzed motif enrichment at the promoter of deregulated genes following *HOXA-AS2* silencing. We found that the binding sites of 53 and 56 transcription factors were enriched at genes that were upregulated in GSC-6 and -11 cells, respectively. Similarly, downregulated genes were the putative targets of 23 and 34 transcription factors in GSC-6 and -11 cells, respectively (Appendix A). Considering the “top 20” transcription factors showing binding motif enrichment at deregulated genes (Figure 5c), we identified 13 transcription factors that were differentially expressed in at least one *HOXA-AS2*-silenced GSC line compared with the control (shNS). Specifically, *E2F1*, *E2F2,* and *E2F8* (with binding motif enrichment in the promoter of downregulated genes) were downregulated in both *HOXA-AS2*-silenced GSC lines. Among the transcription factors with binding site enrichment in the promoter of upregulated genes, members of the Signal Transducer And Activator Of Transcription (STAT) and Interferon Regulatory Factor (IRF) families, and also Activating Transcription Factor 3 *(ATF3*) and *MYC* were upregulated following *HOXA-AS2* silencing (Figure 5d). To determine whether these transcription factors were direct *HOXA-AS2* targets, we analyzed the correlation between their expression level and that of *HOXA-AS2* in glioma using the RNA-seq data of the 134 *IDHwt* glioma samples from the “TCGA cohort”. We did not observe any significant negative correlation for the upregulated transcription factors with binding site enrichment at the promoter of upregulated genes. Conversely, the *HOXA-AS2* expression level positively correlated with *E2F8* and *E2F1* expression, which, therefore, can be considered direct *HOXA-AS2* targets (Figure 5d and Appendix A).

### 2.6. Changes in Gene Expression Are Not Associated with Major Changes in the Chromatin Signature at Deregulated Genes Following HOXA-AS2 Silencing in GSCs

Then, we investigated the molecular bases of the widespread transcriptional alteration observed following *HOXA-AS2* silencing in the two GSC lines. We did not detect any significant DNA methylation change genome-wide and at the promoter of genes deregulated upon *HOXA-AS2* silencing in GSCs as observed in GSC-11 (MethylationEPIC BeadChip) (Figure 6a, Appendix A). Next, we analyzed histone mark signatures relevant for promoter and/or enhancer activity (i.e., the permissive H3K4me3, active H3K27ac, and repressive H3K27me3 histone marks) by ChIP-seq in silenced GSC-6 cells due to the strongest effect of *HOXA-AS2* silencing in this line (Figure 5a). Genome-wide changes were limited. We observed H3K4me3 changes (fold change > 2, FDR < 0.01) in 1133 regions that included 120 promoters and 55 enhancers. On the other hand, we detected H3K27ac and H3K27me3 changes only in 317 (43 promoters/15 enhancers) and 332 (21 promoters and 1 enhancer) regions, respectively (Appendix A). The limited number of putative regulatory regions affected (i.e., enhancers and promoters) can hardly account for the widespread gene deregulation observed following *HOXA-AS2* silencing in GSC-6 cells (Figure 5a, Appendix A). To precisely determine the proportion of genes the deregulation of which was associated with chromatin signature alterations, we investigated the histone mark profile at the promoters of the 1725 genes with expression changes following *HOXA-AS2* knockdown in GSC-6 cells. We detected concomitant changes in gene expression and promoter signature only for very few genes. We observed a gain in H3K4me3 and H3K27ac at the promoter of 15 and 2 genes (1.5% and 0.2% of all upregulated genes), respectively. We did not find any significant change in the repressive H3K27me3 mark (Figure 6b). Consistently, the two upregulated genes with a gain in H3K27ac, Neuralized E3 Ubiquitin Protein Ligase 3 (*NEURL3*) and Interferon Alpha Inducible Protein 27 (*IFI27*), also showed a H3K4me3 gain at their promoter (Figure 6c, Appendix A).

This analysis highlighted that for most genes, changes in gene expression following *HOX-AS2* silencing was not associated with changes in the histone mark signature at their promoter. We obtained similar results (i.e., absence of change in histone mark signatures) also at the promoter of the putative direct targets of *HOXA-AS2* (Figure 5c,d): the downregulated *E2F-8* and *E2F-1* genes and the upregulated *STAT1* gene (Figure 6d,e, Appendix A).

Altogether, these observations suggest that *HOXA-AS2* does not modulate the expression of its target genes through transcriptional regulation.

## 3. Discussion

Here, we investigated the regulation and role of the lncRNA *HOXA-AS2* in GSCs. The Gencode gene project predicts ~12 *HOXA-AS2* isoforms, but there is no consensus in the literature on what are the most relevant isoforms in different tumor types. Many studies focused on one isoform without providing a rationale for this choice. Therefore, first, we used RNA-seq, associated with Race approaches, to show that the longer variant of the ENST0000522193.1 isoform is the major *HOXA-AS2* transcript in glioma samples. This variant contains at least two exons and is 1049bp in length after splicing. This isoform was included in the isoforms analyzed in the first studies on *HOXA-AS2* in cancer (e.g., [20,23]).

Our data confirmed previous observations made in glioma samples classified according to the 2007 WHO criteria (i.e., GBM and lower-grade glioma) that *HOXA-AS2* is upregulated in glioma and that its expression level is positively associated with advanced tumor stages [25,26]. By analyzing glioma samples classified according to the most recent WHO recommendations [3], the present study refined these observations and also extended them to GSC lines. We found that *HOXA-AS2* expression is a characteristic of *IDHwt* glioma samples and GSCs, while it is not expressed in *IDHmut* glioma samples. This is similar to our previous observation that widespread HOX gene overexpression is a molecular signature of both *IDHwt* glioma samples and GSCs [7]. However, the HOX genes expression pattern can differ between *IDHwt* glioma tissue and GSCs. Specifically, we observed that the expression of the lncRNA *HOTAIR*, which plays a critical oncogenic role in malignant glioma [31,32], was restricted to *IDHwt* glioma samples. Conversely, *HOXA-AS2* was expressed in both *IDHwt* glioma samples and GSC lines, suggesting that it may contribute to GSC biology.

In all cancer types where it has been studied, including malignant glioma, *HOXA-AS2* has been found to have oncogenic functions, mainly by promoting proliferation [25,33]. Similarly, our silencing experiments in GSC lines suggest that *HOXA-AS2* influences cell proliferation, by acting primarily on E2F-8 and E2F-1, two main cell cycle regulators the deregulation of which promotes gliomagenesis [34,35]. In addition to promoting the expression of cell cycle genes, *HOXA-AS2* also negatively regulated a subset of genes of the inflammatory pathway in GSCs, although probably in an indirect manner. Motif enrichment analysis suggested that this function could be mediated by the initial downregulation of a few members of the STAT and IRF families and also *ATF3.* This observation is consistent with the documented tumor suppressor role of STAT1 and ATF3 in glioma and GSC [36,37]. The inactivation of *ATF3* is essential for the oncogenic potential of GSCs [37]. In addition, *STAT1* downregulation allows GSCs to evade type I interferon suppression [38]. Altogether, our findings suggest that *HOXA-AS2* can influence, directly or indirectly, several signaling pathways that are instrumental for the GSC oncogenic potential.

Our findings provide new insights into the *HOXA-AS2* mechanism of action by suggesting that its effect in GSCs might rely mainly on post-transcriptional regulation. Its ectopic overexpression did not affect H9-NSC morphology and proliferation, and led to the deregulation of only four genes. Yet, three of them, *FABP3, RGS16,* and *EBF2,* are relevant for glioma biology [28,29,30]. This observation suggests that *HOXA-AS2* overexpression, as detected in *IDHwt* glioma and in GSCs, is not sufficient on its own to promote a pathological phenotype. Given the HOX gene widespread reactivation and their functional role in glioma and GSC [7,8,9], it can be hypothesized that *HOXA-AS2* collaborates with some of them in this process, such as the lncRNA *HOTAIRM1* that is overexpressed in both *IDHwt* samples and GSCs, such as *HOXA-AS2*. Other/additional specific but yet-unknown partner(s) might be required for *HOXA-AS2* oncogenic function in GSCs. Our findings in *HOXA-AS2*-silenced GSC lines suggest that such partner(s) may be implicated in post-transcriptional regulation. Indeed, the vast majority of genes transcriptionally deregulated following *HOXA-AS2* silencing in GSCs did not show any change in relevant histone mark signatures at their promoter, including at genes we identified as direct *HOXA-AS2* targets, suggesting a mechanism that does not rely on transcriptional regulation. It has been proposed that HOXA-AS2 might regulate gene expression by acting as a scaffold for epigenetic modifiers or by sponging miRNAs [26,39]. Our findings are in favor of an effect mediated through miRNA sponging in GSCs. However, it is also important to stress that the silencing strategy used here is not dedicated to reveal an effect mediated by epigenetic modifiers. If *HOXA-AS2* regulates genes by acting as a scaffold, epigenetic marks, such histone modifications, deposited by modifiers could be maintained despite its silencing. Therefore, additional studies are needed to fully understand the relative contribution of *HOXA-AS2* as a scaffold in GSCs, for instance, by evaluating whether *HOXA-AS2* physically interacts with target genomic regions and by identifying its protein partners.

Altogether, our study revealed that *HOXA-AS2* is a key actor of GSC biology. Our findings support a model in which its overexpression triggers a cascade of events that promote, through direct and indirect mechanisms, cell proliferation and immune tolerance. Additional studies are needed to test and validate this model in vivo. Nevertheless, *HOXA-AS2* is a relevant candidate to support the GSC tumorigenic potential.

## 4. Materials and Methods

### 4.1. Biological Material

Tumors, GSC lines, and control brain tissue samples were previously described in [7,40]. Briefly, adult diffuse glioma samples (*n* = 70), resected between 2007 and 2014, were from Clermont-Ferrand University Hospital Center, Clermont-Ferrand, France (“Tumorotheque Auvergne Gliomes”, ethical approval DC-2012-1584). This study was approved by the relevant ethics committees and competent authorities, and the study protocols followed the World Medical Association Declaration of Helsinki. Tumors were classified according to their IDH mutation status: *IDHwt* (*n* = 55) and *IDHmut* (*n* = 15), following the 2021 WHO classification [3]. Fifteen control brain tissue samples (healthy controls; samples removed by autopsy 4–16 h after accidental death) were obtained from the Brain and Tissue Bank of Maryland (mean age: 27.3 years, standard deviation 2 years). These samples, identified by the Brain and Tissue Bank of Maryland as corpus callosum (*n* = 8) and frontal cortex (*n* = 7), correspond to white matter enriched in astrocytes and oligodendrocytes and are relevant non-cancer controls for gliomas. Cell pellets from eight GSC lines (GSC-1, GSC-2, GSC-3, GSC-5, GSC-6, GSC-9, GSC-10, and GSC-11) derived from patients with *IDHwt* GBM were obtained from Poitiers University Hospital Centre, Poitiers, France (Ethical approval DHOS/OPRC/FCnotif-tumoro-jun04: 04056) and were previously characterized [41,42,43]. Expression validation cohorts, obtained from The Cancer Genome Atlas (TCGA) research network, were described in [27]. For this study, *IDHmut* (*n* = 415) and *IDHwt* (*n* = 134) samples with RNA expression (RNA-seq) data were used. The clinical and molecular data of these patients were retrieved from the cBioPortal for Cancer Genomics (https://www.cbioportal.org/ accessed on 17 November 2021) [44,45]. Processed RNA-seq data were obtained from the TCGA website (https://portal.gdc.cancer.gov/ accessed on 17 November 2021) and analyzed as described below. Human neural stem cell pellets (derived from the H9 human embryonic stem cell, hESC, line) were from Invitrogen (N7800-100, Illkrich, France).

### 4.2. Transcript Expression Analysis

#### 4.2.1. RNA Extraction

Total RNA was isolated from frozen tissue samples and frozen cell pellets as previously described [7]. Differential isolation of cytoplasmic and nuclear RNA from GSC samples was performed with the “Cytoplasmic and Nuclear RNA purification Kit” from Norgen Biotek (21000; Thorold, ON, Canada).

#### 4.2.2. RACE-PCR

5′ and 3′ RACE-PCR amplifications were performed as previously described [46] with the GeneRacer Kit from Invitrogen (L150201, Illkrich, France). The primers used are described in Appendix A.

#### 4.2.3. RT-qPCR

RT-qPCR data were from previously performed microfluidic-based qPCR assays using glioma and control brain samples and GSC lines (*n* = 10 control samples, *n* = 8 *IDHmut* and *n* = 43 *IDHwt* glioma samples, and *n* = 6 GSC lines) to assess the expression of 37 coding and 17 noncoding HOX transcripts [7]. The primers used for *HOXA-AS2*, *HOTAIR, HOTAIRM1,* and *HOTIP* amplification are described in Appendix A. The choice of the control housekeeping genes (*PPIA*, *TBP,* and *HPRT1)* was based on Valente et al.’s and Kreth et al.’s studies [47,48]. These three genes display a similar expression level in IDHwt (*n* = 134) and IDHmut (*n* = 415) samples from the TCGA cohort.

#### 4.2.4. RNA-Seq

Strand-oriented RNA-seq analysis of total RNA from tumor and control tissues samples (*n* = 3 brain control, *n* = 8 *IDHwt*, *n* = 5 *IDHmut* GBM samples), two GSC lines (GSC-1 and GSC-2), and also of mRNA from the GSC-6 and GSC-11 lines that express scramble shRNA were previously performed [7,8,9,10,11,12,13,14,15,16,17,18,19,20,21,22,23,24,25,26,27,28,29,30,31,32,33,34,35,36,37,38,39,40] (GSE123892, GSE161438, and GSE161437). For this study, strand-oriented RNA-seq was performed with mRNA from the GSC-6 and GSC-11 lines in which *HOXA-AS2* was silenced (shRNA-2 and shRNA-3) and from H9-NSCs transfected with an empty vector or the *HOXA-AS2* expression vector (two independent transfection experiments). Data were analyzed as in [40]. Briefly, RNA-seq data were mapped to the hg19 human genome assembly using TOPHAT2 (version 2.1.0) (https://ccb.jhu.edu/software/tophat/index.shtml accessed on 22 November 2021) and a transcript annotation file from GENCODE (Release 19). Reads were filtered with SAMTOOLS (v 1.9) (https://github.com/samtools/samtools accessed on 22 November 2021) to keep only properly paired reads. The read count per gene was obtained with the HTseq-count script. Strand-specific RNA-seq coverage was assessed with SAMTOOLS (v 1.9), GENOMECOVERAGEBED (v2.27.1) (https://bedtools.readthedocs.io/en/latest/content/tools/genomecov.html accessed on 22 November 2021), and BEDGRAPHTOBIGWIG (http://hgdownload.soe.ucsc.edu/admin/exe/linux.x86_64/bedGraphToBig-Wig accessed on 22 November 2021) and visualized using the UCSC Genome Browser. Differential expression analyses were performed on read counts using EdgeR packages (https://bioconductor.org/packages/release/bioc/html/edgeR.html accessed on 7 December 2021), R(4.0.2). Before the differential analysis, poorly expressed genes (sum count for all samples < 10) or located on the chromosomes X, Y, and M were excluded. Genes were considered as differentially expressed when |log2(fold change)| > 1 with an adjusted *p* value (FDR) < 0.05. For the differential analysis of the H9-NSC samples, replicates were treated as paired samples. Raw data are accessible at GSE199030.

### 4.3. DNA Methylation Analysis

#### 4.3.1. DNA Extraction

DNA was isolated as previously described [7].

#### 4.3.2. Array-Based DNA Methylation Analysis

Data for tumor, control, and GSC samples were previously obtained using the Human Methylation 450K (HM450K) BeadArray platform [40] (*n* = 55 *IDHwt* glioma, *n* = 8 control brain samples (GSE123678)), and Infinium Methylation EPIC BeadChips platform [7] (GSC-11 cells that express scramble shRNA) (GSE161175). For this work, DNA from H9-NSCs transfected with an empty vector or the *HOXA-AS2* expression plasmid, and DNA from GSC-11 cells that express the two *HOXA-AS2* shRNAs (shRNA-2 and shRNA-3) were also analyzed using Infinium Human Methylation EPIC BeadChips (Illumina). DNA bisulfite conversion and array hybridization were performed by IntegraGen, SA (Evry, France) using the Illumina Infinium HD methylation protocol (Illumina, San Diego, CA, USA). Analyses were performed as previously described [40]. Briefly, β-values were computed using the GenomeStudio software from Illumina. Probes with poor-quality signal, missing signal, overlapping with common SNPs, or present on gonosomes were excluded. Differential analysis was performed using the limma package (https://bioconductor.org/packages/release/bioc/html/limma.html accessed on 11 April 2022) and probes were considered differentially methylated when the adjusted *p* value (FDR) was <0.05 and when the difference between groups was >0.1. In-house R scripts were used to produce bedgraph files to visualize signals on UCSC Genome Browser. Raw data are accessible at GSE199030.

### 4.4. Chromatin Analysis

#### 4.4.1. ChIP qPCR

Anti-H3K9ac (Millipore 06-942, Molsheim, France), -H3K4me3 (Diagenode 03-050, Seraing, Belgium), and -H3K27me3 (Millipore 07-449, Molsheim, France) antibodies were used to quantify these histone marks at the *HOXA-AS2* CpG island/promoter region by ChIP of native chromatin isolated from glioma samples and brain controls, as previously described [49]. The bound/input ratios were calculated and normalized to the precipitation level at the *TBP* promoter for the anti-H3K9ac and -H3K4me3 ChIPs and at the *SP6* promoter for the anti-H3K27me3 ChIP. The primers used are described in Appendix A.

#### 4.4.2. ChIP-Seq of GSC Samples

ChIP-seq data obtained using native chromatin isolated from GSC-6 and GSC-11 cell samples that express scramble shRNAs were previously obtained [7] (GSE161436). New ChIP-seq were performed using native chromatin isolated from GSC-6 cell samples that express *HOXA-AS2* shRNAs (shRNA-3), as previously described [7]. Antibodies against H3K27ac (Abcam Ab4729) (Abcam, Paris, France), H3K4me3 (Diagenode 03-050), and H3K27me3 (Millipore 07-449) were used. Background precipitation levels were determined by performing mock precipitations with a non-specific IgG antiserum (Sigma-Aldrich C2288, Saint-Quentin-Fallavier, France), and experiments were validated by qPCR before sequencing. Library preparation and sequencing on a NovaSeq 6000 instrument (Illumina) were performed by IntegraGen SA, according to the manufacturer’s recommendations (mean of 20 million paired reads per sample). ChIP-seq reads of replicate 1 (R1) were mapped to the human genome (hg19) using BOWTIE2 (v 2.3.4.3) (http://bowtie-bio.sourceforge.net/bowtie2/index.shtml accessed on 13 January 2022). Alignments were filtered according to their quality (Mapq > 30) using Sam-Tools (v 1.9). ChIP-seq signals were generated with Bamcoverage (v 3.1.3) (options: normalizeUsing RPKM, extendReads 200, ignoreDuplicates, binSize 20) and visualized with UCSC Genome Browser. Peak calling and differential peak calling between GSC-6 samples that express scramble and *HOXA-AS2* shRNAs were computed with sicer_df (https://zanglab.github.io/SICER2/ accessed on 18 January 2022). Changes were considered significant only if |log2(fold change)| > 1 and (FDR) < 0.01. Raw data are accessible at GSE199031. ChIP-seq data on H3K4me3 and H3K27me3 in H9-NSC samples were obtained from the Roadmap Epigenomic Project (respectively, GSM772736 and GSM772801).

### 4.5. Functional Annotations

Gene Ontology enrichment analyses were performed with the functional annotation tools in DAVID 6.8 (https://david.ncifcrf.gov/ accessed on 15 December 2021). Regulatory features and cis-regulatory modules were predicted using i-cis Target (https://gbiomed.kuleuven.be/apps/lcb/i-cisTarget/ accessed on 15 December 2021), as previously described (Leboiteux et al., 2021). Only motifs with a normalized enrichment score (NES) above a specified threshold (here 3.0) were considered enriched.

### 4.6. H9-NSC Culture and Transfection

Human Neural Stem Cells (NSC) derived from H9 hESCs (Gibco-Thermo Fisher Scientific, Illkirch, France) were cultured according to the manufacturer’s recommendations (name of maker). Cells were maintained in KnockOut D-EM/F12 (Gibco-Thermo Fisher Scientific) medium supplemented with 2 mM Glutamax^®^ (Gibco-Thermo Fisher Scientific), 20 ng/mL of b-FGF (Peprotech, Neuilly-sur-seine, France), and 20 ng/mL of EGF (Peprotech, France) in T75 flasks coated with Geltrex^®^ (Gibco-Thermo Fisher Scientific) in a humidified incubator at 37 °C and with 5% CO_2_. Culture medium was replaced twice per week.

An empty pcDNA3.1 (+) vector and *HOXA-AS2* expression plasmid (Invitrogen-Thermo Fisher Scientific, Illkrich, France) were transfected with Fugene HD (Promega, Charbonniéres-les-bains, France) according to the manufacturer’s recommendations, and stable clones were selected by adding Geneticin^®^ (Gibco-Thermo Fisher Scientific) in the culture medium (50 μL/mL for 24 h and then 200 μL/mL).

### 4.7. GSC Culture and Transduction

GSCs were cultured at 37 °C as proliferative non-adherent spheres in Neurobasal medium (Life Technologies, Carlsbad, CA, USA) supplemented with B27, N2, and bFGF and EGF at 20 ng/mL (Life Technologies). Culture medium was replaced twice per week and when spheres became large, they were enzymatically dissociated with Accutase (Merck-Millipore, Billerica, MO, USA). The molecular features of GSCs are described in [41,42,43], and the self-renewal, differentiation in vitro, and in vivo tumorigenicity (intracranial xenografts in immunodeficient mice) of GSC cultures were evaluated. SMART lentiviral vectors harboring nontargeting shRNA (VSC11713) or *HOXA-AS2* shRNAs (V3SH11249; shRNA-2 V3SH11246-245316977, shRNA-3 V3SH11246-245298993) were purchased from GE Healthcare (Little Chalfont, UK). Sequences are in Appendix A. Viral particles were produced and concentrated by the Vectorology platform of Montpellier (Biocampus, Montpellier, France). GSCs were infected using a multiplicity of infection of 10 and processed 5 days later for RNA-seq, proteomic, DNA methylation, and ChIP-seq analyses.

### 4.8. Cellular Analyses

#### 4.8.1. Cell Proliferation Assay

Quantification of cell proliferation was performed with the Cell proliferation Kit II XTT (Merck, Quentin Fallavier, France) and CellTiter 96^®^ Aqueous Non-Radioactive Cell Proliferation Assay (MTS) (Promega, Charbonniéres-les-bains, France) for H9-NSC and GSCs, respectively, according to the manufacturer’s recommendations. For GSC, quantification was performed 2, 5, 7, and 9 days after infection with shRNA.

#### 4.8.2. Apoptosis Assay

Detection of apoptotic cells in H9-NSC was performed with the ApopTag^®^ Red In Situ Apoptosis Detection Kit (Merck, France), according to the manufacturer’s recommendations. Total and apoptotic cells were quantified under a fluorescence microscope, using DAPI for staining of nuclei.

## Figures and Tables

**Figure 1 ijms-23-04743-f001:**
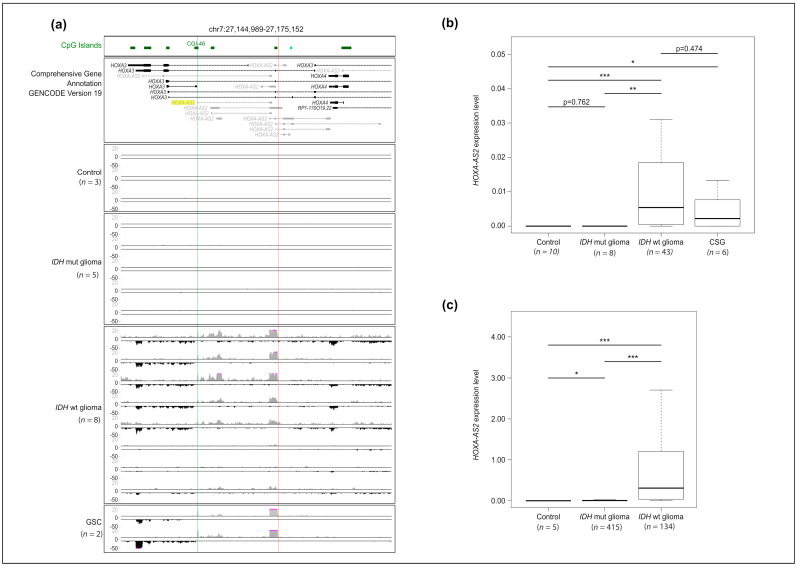
Expression of the *HOXA-AS2* transcript in *IDHwt* glioma samples and GSC lines. (**a**) Strand-oriented RNA-seq signals along the genomic region encompassing *HOXA-AS2* isoforms in control (*n* = 3), *IDHmut* (*n* = 5) and *IDHwt* (*n* = 8) glioma, and Glioma Stem Cell (GSC) (*n* = 2) samples. For each sample, sense (in black) and antisense (in gray) transcription signals are shown in the lower and upper panels, respectively. The positions of 5′ and 3′ ends, identified using the RACE-PCR approach, in one *IDHwt* glioma sample are shown by a green and red vertical line, respectively. (**b**) and (**c**) Relative expression level of *HOXA-AS2* in control, *IDHmut* and *IDHwt* glioma, and GSC samples from our cohort analyzed by microfluidic-based RT-qPCR (**b**) and from the TCGA cohort, analyzed by RNA-seq (**c**). In (**b**), values are the fold change relative to the geometrical mean of the expression of the housekeeping genes *PPIA*, *TBP*, and *HPRT1.** *p* < 0.05, ** *p* < 0.01, *** *p* < 0.001 (Mann-Whitney *U*-test).

**Figure 2 ijms-23-04743-f002:**
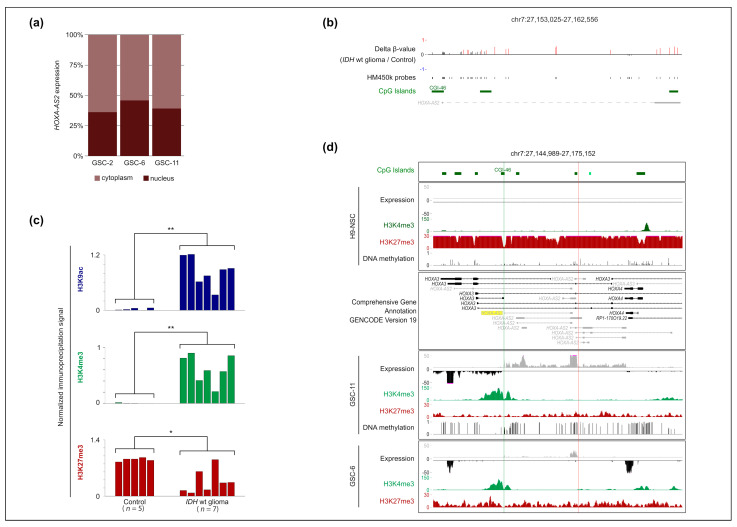
Subcellular distribution and molecular profile of HOXA-AS2. (**a**) Relative distribution of HOXA-AS2 transcripts in cytoplasm and nucleus of three GSC lines. Expression levels were normalized to U6 and 18S expression in the cytoplasmic and nuclear fractions, respectively. (**b**) DNA methylation changes (compared with control samples; *n* = 5) along the HOXA-AS2 genomic region in IDHwt (*n* = 55) glioma samples, detected with the HM450K array. Significantly hypermethylated regions are in red (no hypomethylated region was observed). * *p* < 0.05, ** *p* < 0.01 (Mann-Whitney *U*-test). (**c**) ChIP analysis of H3K9ac, H3K4me3, and H3K27me3 at the HOXA-AS2 promoter region in control brain samples (*n* = 5), and IDHmut (*n* = 5) and IDHwt (*n* = 7) glioma samples. The values obtained for each sample are shown. The precipitation level was normalized to that at the TBP promoter (for H3K4me3 and H3K9ac) and at the SP6 promoter (for H3K27me3). (**d**) Genome Browser integrative view at the HOXA-AS2 locus to show H3K4me3, H3K27ac, and H3K27me3 enrichment, DNA methylation (in the GSC-11 and H9-NSC lines), and the strand-oriented RNA-seq signal in GSC-6, GSC-11, and H9-NSC lines. The positions of the 5′ and 3′ ends identified by RACE-PCR are shown by a green and red line, respectively.

**Figure 3 ijms-23-04743-f003:**
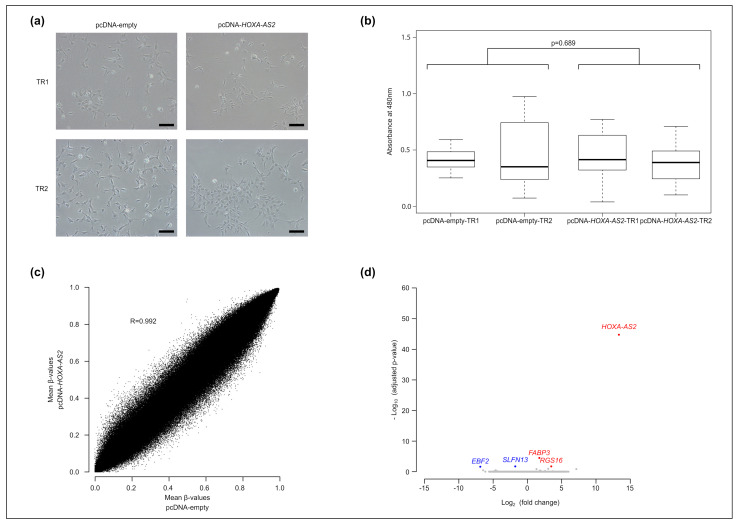
*HOXA-AS2* overexpression in the H9-NSC line. (**a**) Representative images of the morphology of *HOXA-AS2*-expressing (right panels; *n* = 2) and control (left panels, *n* = 2) H9-NSC lines. These cell lines were obtained from two independent transfections, referred to as TR1 and TR2, respectively. (**b**) Colorimetric assay (*n* = 5) to quantify cell proliferation in the two *HOXA-AS2*-expressing H9-NSC lines and their respective control line. (**c**) Infinium Methylation EPIC BeadChip data were used to produce scatter plots to correlate DNA methylation patterns of *HOXA-AS2*-expressing (*n* = 2) and control (*n* = 2) H9-NSC lines. (**d**) Volcano plot of differential gene expression in *HOXA-AS2*-expressing (*n* = 2) and control (*n* = 2) H9-NSC lines.

**Figure 4 ijms-23-04743-f004:**
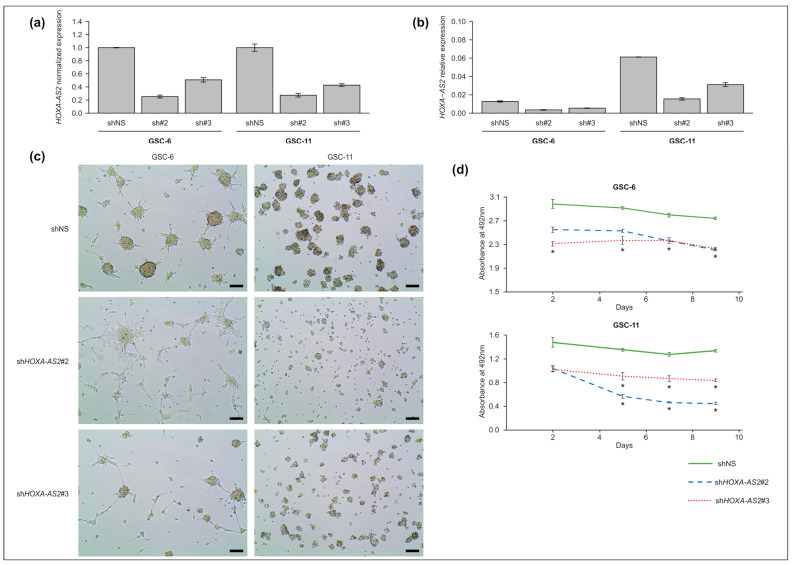
*HOXA-AS2* knockdown in GSC lines. (**a**,**b**) Relative expression level of *HOXA-AS2* in GSC-6 and GSC-11 cells five days after transduction with lentiviruses expressing two different *HOXA-AS2* shRNAs (sh#2 and sh#3) or with a nonsilencing shRNA used as control (shNS). Values are the fold change relative to the geometrical mean of expression of the housekeeping genes *PPIA*, *TBP*, and *HPRT1.* Data are presented as relative expression normalized (**a**) and not normalized (**b**) to the expression level in shNS cells. (**c**) Representative images of neurosphere formation in GSC-11 and GSC-6 cells five days after transduction with lentiviruses expressing two different shRNAs (shHOXA-AS2#2, #3) or with a nonsilencing shRNA used as control (shNS). Scale bar = 100 μm. (**d**) Growth curves of GSC-6 and GSC-11 cells after *HOXA-AS2* silencing or not at days 2, 4, 7, and 9 after transduction; * *p* < 0.05 (Mann–Whitney test).

**Figure 5 ijms-23-04743-f005:**
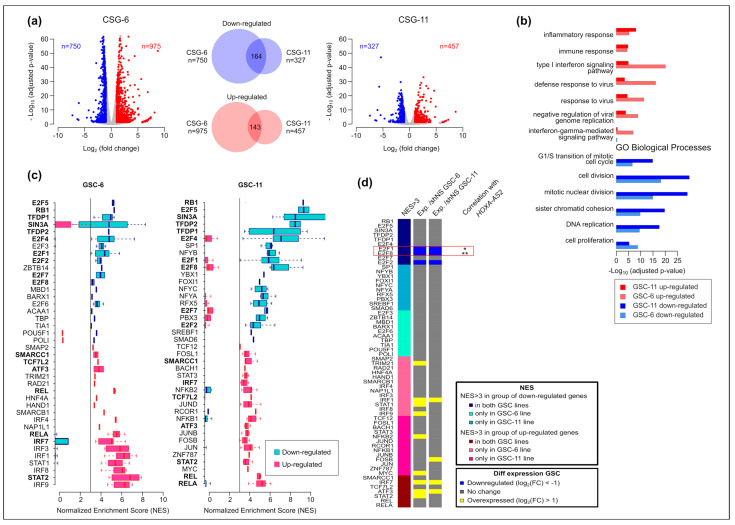
Identification of *HOXA-AS* putative direct targets in GSCs. (**a**) Volcano plots analysis of differential gene expression following *HOXA-AS2* silencing in GSG-6 (left) and GSC-11 cells (right). Blue and red dots represent genes that were significantly down- or upregulated, respectively. Venn diagrams (middle) showing the overlapping between downregulated (blue) and upregulated (red) genes. (**b**) Gene Ontology terms (biological processes) enriched in downregulated (in blue) and upregulated (in red) genes in GSC-11 and GSS-6 cells following *HOXA-AS2* silencing. The five highest terms of each category, in each GSC line, are shown. (**c**) Transcription factor motif enrichment at the promoter (defined as the area covering -1 kbp of RefSeq TSS) of up- and downregulated genes following *HOXA-AS2* silencing in GSG-6 and GSC-11 cells, calculated using i-cis Target and represented as normalized enrichment score (NES). Transcription factors with motif enrichment in both cell lines are in bold. When a transcription factor harbors several binding motifs, data are presented as a box plot. Only the top 20 transcription factors with NES >3 are shown. (**d**) Expression changes, assessed using publicly available RNA-seq data from 134 *IDHwt* glioma samples (TCGA cohort), of the transcription factors identified in (**c**) and that were deregulated or upon *HOXA-AS2* silencing compared with control (shNS) in GSC-6 and -11 cells (blue, downregulated: log2(Fold Change) < −1; gray, no change; yellow, overexpressed: log2(Fold Change) > 1). Significant correlation with *HOXA-AS2* expression, assessed for the 13 up- or downregulated transcription factors, is indicated in the right column. * *p* < 0.05, ** *p* < 0.01 (Mann-Whitney *U*-test).

**Figure 6 ijms-23-04743-f006:**
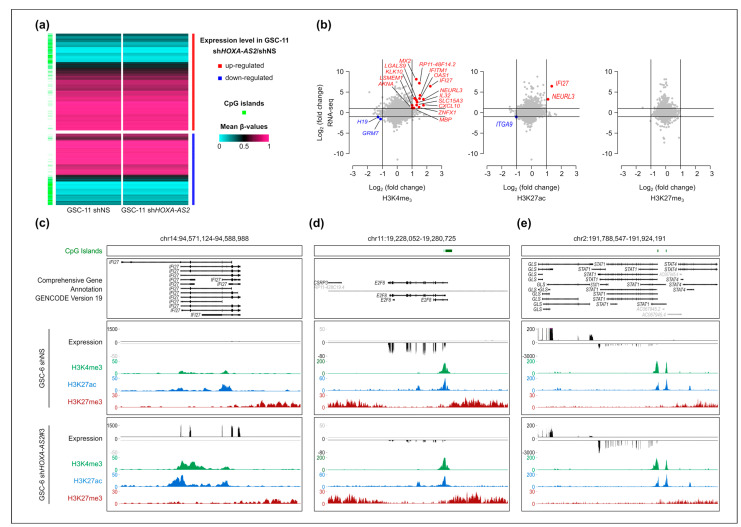
DNA methylation and histone mark changes at deregulated genes following *HOXA-AS2* knockdown. (**a**) Promoter DNA methylation level (mean β-values) at the 784 genes deregulated (457 up- and 327 downregulated) following *HOXA-AS2* knockdown assessed in control GSC-11 cells (shNS) (*n* = 2) and in silenced cells *(Sh-HOXA-AS2)* (*n* = 4). (**b**) Integrative analysis of changes in gene expression (Log2 FC) and in histone mark enrichment following *HOXA-AS2* silencing in GSC-6 cells. Genes with significant changes in gene expression and histone mark profile (Log2 FC > 1 or < − 1; FDR < 0.05) are shown in red (upregulated) and blue (downregulated). (**c**–**e**) Genome Browser view at the *IFI27* (**c**), *E2F8* (**d**), and *STAT1* (**e**) loci to show H3K4me3, H3K27ac, and H3K27me3 enrichment, and the strand-oriented RNA-seq signal in GSC-6 cells transduced with lentiviruses expressing shNS or shHOXA-AS2#3.

## Data Availability

Data supporting the reported results can be found at NCBI Gene Expression Omnibus (GEO; https://www.ncbi.nlm.nih.gov/geo/ accessed on 15 April 2022) under the following accession numbers (see corresponding material and method section for details): GSE199030, GSE123892, GSE161438, and GSE161437 for RNA-seq data. GSE199032 and GSE123678 for the Epic and HM450K platforms DNA methylation data. GSE199032, GSE161436, GSM772736, and GSM772801 for ChIP-seq data.

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
