# Peer review of "The Long Non-Coding RNA HOXA-AS2 Promotes Proliferation of Glioma Stem Cells and Modulates Their Inflammation Pathway Mainly through Post-Transcriptional Regulation"

_ijms, 2022, doi:10.3390/ijms23094743_

Round 1

Reviewer 1 Report

The work is well-designed; the hypothesis, results, and conclusions are well-defined. The manuscript explores the hypothesis that the long non-coding RNA HOXA-AS2 has a significant impact on GSC biology using descriptive and functional analyses of glioma samples, cell lines, and RNA-seq and methylome integrative analyses combined with non-cording RNA HOXA-AS2 overexpression and knockdown. Authors revealed that the E2F8, E2F1, STAT1, and ATF3 transcriptional factors are the key responsive transducers of the HOXA-AS2 in gliomas. The unexplored hypothesis that the HOXA-AS2 might regulate gene expression by miRNA sponging is interesting and could be the logistic continuation of the current work. I recommend the manuscript for publication in the International Journal of Molecular Sciences with minor revision.  The minor suggestions are attached:

  1. Line 126. Please explain the choice of the housekeeping genes like PPIA, TBP, and HPRT1.
  2. Please include the expression of the housekeeping gene (genes) for the TCGA cohort (as a graph or as a statement) for Fig. 1C.
  3. Please provide overall survival data (Kaplan-Meier curves) for patients from the TCGA cohort (IDH1 wild type) harboring high and low expression levels of HOXA-AS2 transcripts.
  4. Line 471. Delete extra space between words.
  5. Please be aware that the R2: (Genomic Analysis and Visualization platform) provides data related to the HOXA-AS2 transcript expression in the wild-type and IDH1 mutant glioma samples (including Kaplan-Meier curves). 

Author Response

We thank the reviewer for its constructive comments on the manuscript. We have read these comments carefully and are now providing a point-by-point response.

  1. Line 126. Please explain the choice of the housekeeping genes like PPIA, TBP, and HPRT1.
  2. Please include the expression of the housekeeping gene (genes) for the TCGA cohort (as a graph or as a statement) for Fig. 1C.

Response to point 1 & 2: For the choice of suitable housekeeping genes we referred to two studies that were dedicated to the identification of control genes for expression analysis in human glioma (Valente at al., BMC Mol Biol 2009; Kreth et al., Neuro Oncol 2010). Among  the HK control genes proposed in these two studies we have retained PPIA, TBP, and HPRT1 that we observed to have a similar expression level (assessed by RNA-seq) between IDwt (n=134) and IDHmut (n=415) samples of the TCGA cohort.

We have added this information in the material and method section and include the two studies in the reference list.

  1. Please provide overall survival data (Kaplan-Meier curves) for patients from the TCGA cohort (IDH1 wild type) harboring high and low expression levels of HOXA-AS2 transcripts.
  2. Please be aware that the R2: (Genomic Analysis and Visualization platform) provides data related to the HOXA-AS2 transcript expression in the wild-type and IDH1 mutant glioma samples (including Kaplan-Meier curves). 

Response to point 3 & 5: We have now did this analysis by using data from the GDC data portal (TCGA cohort) and R2 platform  (“French”  cohort).  This analysis show that High HOXA-AS2 expression tend to correlate with a poorer survival  outcome among IDHwt samples.

We have added this information in the text  and  presented it in an new supplementary figure  (Supp Figure 2).

  1. Line 471. Delete extra space between words.

Response to point 4: Done

Reviewer 2 Report

In this manuscript, Le Boiteux et al. characterized the lncRNA HOXA-as2expression in in glioma samples, particularly in primary GBM and in GSC lines, differentiating the most aggressive glioma IDHwt, where HoxA-as2 is overexpressed, from IDHmut glioma with better prognosis and without expression of this lncRNA. The overexpression of Hoxa-as2 in normal H9-NSC line did not lead to detectable phenotypic alterations and relevant differential gene expression pattern. Silencing of HoxA-as2 in two GSC line decreased cell proliferation and altered expression of several hundred of genes. By integrative analysis the authors demonstrate that the changes in gene expression following HoxA-as2 silencing, are not associated to DNA methylation changes of the deregulated genes, suggesting a post-transcriptional regulation.

Moreover, the authors identify E2F8, E2F1, and STAT1 as direct target modulated by HoxA-as2, but not mediated by transcriptional regulation mechanisms.

The hypotheses of the research are clear, scientifically justified with appropriate controls and do not raise any objections. The work methodology described in the main body of the paper, as well as in the supplementary materials, is written in a clear, straightforward manner, giving all the required details.
My comments are as follows:

Major:

-From www.ensemble.org site, results that ENST00000522193.1 transcript cited in the manuscript is not among the longer isoform of the gene HoxA-as2 (ENSG00000253552), but among the smaller. Its length is 339nt. (http://www.ensembl.org/Homo_sapiens/Transcript/Summary?db=core;g=ENSG00000253552;r=7:27113936-27122199;t=ENST00000522193).

-In Supplementary fig.1a: strand-oriented RNA-seq signals along HOXA-as2 region in control (n=3) and IDHmut (n=5) samples, described in figure legend, are missing.

-To verify the over-expression of HOXA-as2 in H9-NSC lines, the authors show in Suppl. Fig2A the fold change relative to the geometrical mean expression of three housekeeping genes, analyzed by RT-PCR, but it may be more useful to show the fold change relative to HOXA-as2 endogenous level.

-line 272; in the legend of figure 5b the authors declare The five highest terms of each category are shown but in the picture are shown seven category for up-regulated and six for down-regulated terms.

- the paragraph 2.6 is confusing; the authors investigated the transcriptional alteration observed following HOXA-as2 silencing in the two GSC lines, but in fig 6a and in suppl Fig4a are showed only data from GSC-11.;

-and then in Line 313: Fig 5E : does not exist in the manuscript;

- and Line 314: the downregulated E2F-8 and E2F-1 genes 313 and the upregulated STAT1 gene (Fig 6D, Supp Fig 4D) : the reference in figure for STAT1 is missing .

Author Response

We thank the reviewer for its constructive comments on the manuscript. We have read these comments carefully and are now providing a point-by-point response.

1- From www.ensemble.org site, results that ENST00000522193.1 transcript cited in the manuscript is not among the longer isoform of the gene HoxA-as2 (ENSG00000253552), but among the smaller. Its length is 339nt. (http://www.ensembl.org/Homo_sapiens/Transcript/Summary?db=core;g=ENSG00000253552;r=7:27113936-27122199;t=ENST00000522193).

Response to point 1: We agree with the reviewer that ENST00000522193.1 is among the shorter from of HOXA-AS2 isoforms, with 339nt in length. By “longer variant of the ENST0000522193.1 isoform” we mean that the major HOXA-AS2 transcript we identified in glioma corresponds to the ENST0000522193.1 isoform but with a longer exon2 than those described in the database (Please see fig1a, between the green and red lines).

2- In Supplementary fig.1a: strand-oriented RNA-seq signals along HOXA-as2 region in control (n=3) and IDHmut (n=5) samples, described in figure legend, are missing.

Response to point 2: Sorry for this typo, we have corrected it.

3-To verify the over-expression of HOXA-as2 in H9-NSC lines, the authors show in Suppl. Fig2A the fold change relative to the geometrical mean expression of three housekeeping genes, analyzed by RT-PCR, but it may be more useful to show the fold change relative to HOXA-as2 endogenous level.

Response to point 3: Native H9-NSC do not expressed HOXA-AS2 and we thus cannot assess, here, change in expression relative to the endogenous level. We thus present expression relative to expression of housekeeping genes, as is the most suitable way to assess the fold change in such case.

4-line 272; in the legend of figure 5b the authors declare The five highest terms of each category are shown but in the picture are shown seven category for up-regulated and six for down-regulated terms.

Response to point 4: In figure 5b, the five highest terms of each category, and in each GSC lines, are shown. However, these five terms are not all the same between the two cell line. For instance, among the up-regulated genes, three of the five highest term overlap between GSC-6 and GSG-11, leading to show 7 GO term in total. Similarly, four of the five highest term overlap between GSC-6 and GSG-11 among the down-regulated genes.

To improve this point, we have added in the figure legend that “The five highest terms of each category, in each GSC line, are shown.”

5- the paragraph 2.6 is confusing; the authors investigated the transcriptional alteration observed following HOXA-as2 silencing in the two GSC lines, but in fig 6a and in suppl Fig4a are showed only data from GSC-11.;

Response to point 5: We thank the reviewer to have pointed out this typo.  DNA methylation analysis, showed in Fig 6a and sup fig 4a, are showed only for GSC-11. This was indicated in the figure legends but, we agree, was not clear in the result text. We have edit it accordingly.

6- and then in Line 313: Fig 5E : does not exist in the manuscript and Line 314: the downregulated E2F-8 and E2F-1 genes 313 and the upregulated STAT1 gene (Fig 6D, Supp Fig 4D) : the reference in figure for STAT1 is missing

Response to point 6: Here again, we thank the reviewer to have pointed out these typo that we have now corrected. Please read “Fig 5d”, rather than “Fig 5e” and “Fig 6 d,e” rather than “Fig 6d”; 

Round 2

Reviewer 2 Report

In my opinion, the manuscript is now acceptable for the publication in IJMS.